# Learning a Latent Simplex in Input-Sparsity Time

**Ainesh Bakshi**
Carnegie Mellon University
abakshi@cs.cmu.edu

**Chiranjib Bhattacharyya**
Indian Institute of Science
chiru@iisc.ac.in

**Ravi Kannan**
Microsoft Research India
kannan@microsoft.com

**David P. Woodruff**
Carnegie Mellon University
dwoodruf@cs.cmu.edu

**Samson Zhou**
Carnegie Mellon University
samsonzhou@gmail.com

## Abstract

We consider the problem of learning a latent $k$-vertex simplex $\mathcal{K} \subset \mathbb{R}^d$, given access to $\mathbf{A} \in \mathbb{R}^{d \times n}$, which can be viewed as a data matrix with $n$ points that are obtained by randomly perturbing latent points in the simplex $\mathcal{K}$ (potentially beyond $\mathcal{K}$). A large class of latent variable models, such as adversarial clustering, mixed membership stochastic block models, and topic models can be cast as learning a latent simplex. Bhattacharyya and Kannan (SODA, 2020) give an algorithm for learning such a latent simplex in time roughly $O(k \cdot \mathsf{nnz}(\mathbf{A}))$, where $\mathsf{nnz}(\mathbf{A})$ is the number of non-zeros in $\mathbf{A}$. We show that the dependence on $k$ in the running time is unnecessary given a natural assumption about the mass of the top $k$ singular values of $\mathbf{A}$, which holds in many of these applications. Further, we show this assumption is necessary, as otherwise an algorithm for learning a latent simplex would imply an algorithmic breakthrough for spectral low rank approximation.

At a high level, Bhattacharyya and Kannan provide an adaptive algorithm that makes $k$ matrix-vector product queries to $\mathbf{A}$ and each query is a function of all queries preceding it. Since each matrix-vector product requires $\mathsf{nnz}(\mathbf{A})$ time, their overall running time appears unavoidable. Instead, we obtain a low-rank approximation to $\mathbf{A}$ in input-sparsity time and show that the column space thus obtained has small $\sin \Theta$ (angular) distance to the right top-$k$ singular space of $\mathbf{A}$. Our algorithm then selects $k$ points in the low-rank subspace with the largest inner product (in absolute value) with $k$ carefully chosen random vectors. By working in the low-rank subspace, we avoid reading the entire matrix in each iteration and thus circumvent the $\Theta(k \cdot \mathsf{nnz}(\mathbf{A}))$ running time.

## 1 Introduction

We study the problem of learning $k$ vertices $\mathbf{M}_{*,1}, \ldots, \mathbf{M}_{*,k}$ of a latent $k$-dimensional simplex $\mathcal{K}$ in $\mathbb{R}^d$ using $n$ data points generated from $\mathcal{K}$ and then possibly perturbed by a stochastic, deterministic, or adversarial source before given to the algorithm. In particular, the resulting points observed as input data could be heavily perturbed so that the initial points may no longer be discernible or they could be outside the simplex $\mathcal{K}$. Recent work of Bhattacharyya & Kannan (2020) unifies several stochastic models for unsupervised learning problems, including $k$-means clustering, topic models (Blei, 2012), and mixed membership stochastic block models (Airoldi et al., 2014) under the problem of learning a latent simplex. In general, identifying the latent simplex can be computationally intractable. However many special applications do not require the full generality. For example, in a mixture model like Gaussian mixtures, the data is assumed to be generated from a convex combination of density functions. Thus, it may be possible to efficiently approximately learn the latent simplex given certain distributional properties in these models.

Indeed, Bhattacharyya & Kannan (2020) showed that given certain reasonable geometric assumptions that are typically satisfied for real-world instances of Latent Dirichlet Allocation, Stochastic Block

Models and Clustering, there exists an $\widetilde{O}(k \cdot \mathsf{nnz}(\mathbf{A}))$ [1] time algorithm for recovering the vertices of the underlying simplex. We show that, given an additional natural assumption, we can remove the dependency on $k$ and obtain a true input sparsity time algorithm. We begin by defining the model along with our new assumption:

**Definition 1.1** (Latent Simplex Model). Let $\mathbf{M}_{*,1}, \mathbf{M}_{*,2}, \ldots, \mathbf{M}_{*,k} \in \mathbb{R}^d$ denote the vertices of a $k$-simplex, $\mathcal{K}$. Let $\mathbf{P}_{*,1}, \mathbf{P}_{*,2} \ldots \mathbf{P}_{*,n} \in \mathbb{R}^d$ be $n$ points in the convex hull of $\mathcal{K}$. Given $\sigma > 0$, we observe $n$ points $\mathbf{A}_{*,1}, \mathbf{A}_{*,2} \ldots \mathbf{A}_{*,n} \in \mathbb{R}^d$ such that $\|\mathbf{A} - \mathbf{P}\|_2 \le \sigma\sqrt{n}$. Further, we make the following assumptions on the data generation process:

1. **Well-Separateness.** For all $\ell \in [k]$, $\mathbf{M}_{*,\ell}$ has non-trivial mass in the orthogonal complement of the span of the remaining vectors, i.e., for all $\ell \in [k]$, $|\mathrm{Proj}(\mathbf{M}_{*,\ell}, \mathrm{Null}(\mathbf{M} \setminus \mathbf{M}_{*,\ell}))| \ge \alpha \max_\ell \|\mathbf{M}_{*,\ell}\|_2$ where $\mathrm{Proj}(x, U)$ denotes the orthogonal projection of $x$ to the subspace $U$.

2. **Proximate Latent Points.** For all $\ell \in [k]$, there exists a set $\mathcal{S}_\ell \subseteq [n]$ such that $|\mathcal{S}_\ell| \ge \delta n$ and for all $j \in \mathcal{S}_\ell$, $\|\mathbf{M}_{*,\ell} - \mathbf{P}_{*,j}\|_2 \le 4\sigma/\delta$.

3. **Spectrally Bounded Perturbation.** The spectrum of $\mathbf{A} - \mathbf{P}$ is bounded, i.e., for a sufficiently large constant $c$, $\sigma/\sqrt{\delta} \le \alpha^2 \min_\ell \|\mathbf{M}_{*,\ell}\|_2 / ck^9$.

4. **Significant Singular Values.** Let $\mathbf{A} = \sum_{i \in [d]} \sigma_i u_i v_i^T$ be the singular value decomposition and let $0 < \phi \le \mathsf{nnz}(\mathbf{A})/(n \cdot \mathrm{poly}(k))$. We assume that for all $i \in [k]$, $\sigma_i > \phi \cdot \sigma_{k+1}$ and $\|\mathbf{A} - \mathbf{A}_k\|_F^2 \le \phi\|\mathbf{A} - \mathbf{A}_k\|_2^2$.

These assumptions are natural across many interesting applications; see Section 2 for more details. Bhattacharyya & Kannan (2020) introduced the Well-Separateness (1), Proximate Latent Points (2) and Spectrally Bounded Perturbation (3) assumptions. We include an additional Significant Singular Values assumption (4), which is crucial for obtaining a faster running time; we discuss this in more detail below. Our main algorithmic result can then be stated as follows:

**Theorem 1.2** (Learning a Latent Simplex in Input-Sparsity Time). *Given $k \ge 2$ and $\mathbf{A} \in \mathbb{R}^{d \times n}$ from the Latent Simplex Model (Definition 1.1), there exists an algorithm that runs in $\widetilde{O}\left(\mathsf{nnz}(\mathbf{A}) + (n + d)\mathrm{poly}(k/\phi)\right)$ time to output subsets $\mathbf{A}_{\mathcal{R}_1}, \ldots, \mathbf{A}_{\mathcal{R}_k}$ such that upon permuting the columns of $\mathbf{M}$, with probability at least $1 - 1/\Omega(\sqrt{k})$, for all $\ell \in [k]$, we have $\|\mathbf{A}_{\mathcal{R}_\ell} - \mathbf{M}_{*,\ell}\|_2 \le 300k^4\sigma/(\alpha\sqrt{\delta})$.*

Our result implies faster algorithms for various stochastic models that can be formulated as special cases of the Latent Simplex Model, including Latent Dirichlet Allocation for Topic Modeling, Mixed Membership Stochastic Block Models and Adversarial Clustering. We summarize the connections to these applications below. We describe our algorithm and provide an outline to our analysis; we defer all formal proofs to the supplementary material.

## 2 CONNECTION TO STOCHASTIC MODELS

We first formalize the connection between the Latent Simplex Model and numerous stochastic models. In particular, we show that topic models like Latent Dirichlet Allocation (LDA) and Stochastic Block Models can be viewed as special cases of the Latent Simplex Model; we defer discussion on Adversarial Clustering to the supplementary material.

### 2.1 TOPIC MODELS

Probabilistic Topic Models attempt to identify abstract topics in a collection of documents by discovering latent semantic structure (Blei & Jordan, 2003; Blei & Lafferty, 2006; Hoffman et al., 2010; Zhu et al., 2012; Blei, 2012). Each document in the corpus is represented by a bag-of-words vectorization with the corresponding word frequencies. The standard statistical assumption is that the generative process for the corpus is a joint probability distribution over both the observed and hidden random variables. The hidden random variables can be interpreted as representative documents for each topic. The goal is to then design algorithms that can learn the underlying topics. The topics can

---

[1]Throughout the paper we use the notation $\widetilde{O}$ to suppress poly-logarithmic factors.

be viewed geometrically as $k$ latent vectors $\mathbf{M}_{*,1}, \mathbf{M}_{*,2}, \ldots, \mathbf{M}_{*,k} \in \mathbb{R}^d$, where $d$ is the size of the dictionary and $\mathbf{M}_{i,\ell}$ is the expected frequency of word $i$ in topic $\ell$. Since each vector $\mathbf{M}_{*,\ell}$ represents a probability distribution, $\sum_i \mathbf{M}_{i,\ell} = 1$. Let $\mathbf{M}$ be the corresponding $d \times k$ matrix. One important stochastic model is Latent Dirichlet Allocation (LDA) (Blei et al., 2003), where each document consists of $m$ words is generated as follows :

- For all $\ell \in [k]$, we pick topic weights $\mathbf{W}_{j,\ell} \sim \mathsf{Dir}(1/k)$, where $\mathsf{Dir}(1/k)$ is the Dirichlet distribution over the unit simplex. The topic distribution of document $j$ is decided by the topic weights, $\mathbf{W}_{j,\ell}$, and given by $\mathbf{P}_{*,j} = \sum_{\ell \in [k]} \mathbf{W}_{j,\ell} \cdot \mathbf{M}_{*,\ell}$, where $\mathbf{P}_{*,j}$ are latent points.

- We then generate the $j$-th document with $m$ words by taking i.i.d. samples from $\mathsf{Mult}(\mathbf{P}_{*,j})$, the multinomial distribution with $\mathbf{P}_{*,j}$ as the probability vector. The resulting document observed is denoted by the vector $\mathbf{A}_{*,j}$, where for all $i \in [d]$ $\mathbf{A}_{i,j} = \frac{1}{m} \sum_{t=1}^{m} \mathbf{X}_{ij}^{(t)}$,, such that $\mathbf{X}_{ij}^{(t)} \sim \mathsf{Bern}(\mathbf{P}_{ij})$, where $\mathbf{X}_{ij}^{(t)} = 1$ if the $i$-th word was chosen in the $t$-th draw while generating the $j$-th document, and 0 otherwise.

The data generation process of LDA can be viewed as a special case of the Latent Simplex Model, where the $j$-th document is the data point $\mathbf{A}_{*,j}$ generated from the stochastic vector $\mathbf{P}_{*,j}$, a point in the simplex $\mathcal{K}$. The vertices of the simplex are the $k$ topic vectors $\mathbf{M}_{*,1}, \ldots, \mathbf{M}_{*,k}$; the goal is then to recover the vertices of $\mathcal{K}$. We formally justify our assumptions below.

**Lemma 2.1** (LDA as a Latent Simplex). *Given $\mathbf{A}, \mathbf{P}, \mathbf{M}$ following the LDA model as described above, such that for all $\ell \in [k]$, $\|\mathbf{M}_{*,\ell}\|_2 = \Omega(1)$, $m, n = \Omega(poly(k/\alpha))$ and $\delta = c\sigma/\sqrt{k}$, assumptions (2),(3) and (4) from Definition 1.1 are satisfied with high probability.*

### 2.2 MIXED MEMBERSHIP STOCHASTIC BLOCK MODELS

The Stochastic Block Model (Airoldi et al., 2008; Miller et al., 2009; Xing et al., 2010; Fu et al., 2009; Li et al., 2016; Fan et al., 2016) is a well-studied stochastic model for generating random graphs, where the vertices are partitioned into $k$ communities and edges within each community are more likely to occur than edges across communities. Given communities $C_1, C_2, \ldots C_k$, there exists a $k \times k$ symmetric latent matrix $\mathbf{B}$, where, $\mathbf{B}_{\ell_1, \ell_2}$ is the probability that there exists an edge between vertices in $C_{\ell_1}$ and $C_{\ell_2}$. The MMBM can be formalized as the following stochastic process:

- For $j \in [n]$, vertex $j$ picks a probability vector $\mathbf{W}_{*,j} \in \mathbb{R}^k$ representing community membership probabilities that sum to 1, i.e., $\mathbf{W}_{i,j} \sim \mathsf{Dir}(1/k)$ for all $i \in [k]$.
- For all pairs $(j_1, j_2) \in [n]$, vertex $j_1$ picks a community $\ell_1$ proportional to $\mathsf{Mult}(\mathbf{W}_{*,j_1})$ and $j_2$ picks a community $\ell_2$ proportional to $\mathsf{Mult}(\mathbf{W}_{*,j_2})$. The edge $(j_1, j_2)$ is included in the graph with probability $\mathbf{B}_{\ell_1, \ell_2}$. Since $\sum_{\ell_1, \ell_2} \mathbf{W}_{\ell_1, j_1} \mathbf{B}_{\ell_1, \ell_2} \mathbf{W}_{\ell_2, j_2}$ represents the edge probability of the edge $(j_1, j_2)$, the latent variable matrix $\mathbf{P}$ of edge probabilities can be represented as $\mathbf{P} = \mathbf{W}^T \mathbf{B} \mathbf{W}^T$.

However, our reduction is not straightforward since now $\mathbf{P}$ depends quadratically on $\mathbf{W}$ and the only polynomial time algorithms for $\mathbf{B}$ directly rely on semidefinite programming. Further, they require non-degeneracy assumptions in order to compute a tensor decomposition provably in polynomial time (Anandkumar et al., 2014; Hopkins & Steurer, 2017). However, we can pose the problem of recovery of the $k$ underlying communities differently and first pick at random a subset $V_1 \subset [n]$ of $d$ vertices and represent the $\ell$-th community by a $d$-dimensional vector that represents the probabilities of vertices in $[n] \setminus V_1$ belonging to community $\ell$ and having an edge with each of the $d$ vertices in $V_1$. We now define $\mathbf{W}_{(1)}$ to be a $k \times d$ matrix representing the fractional membership of weights of vertices in $V_1$ and $\mathbf{W}_{(2)}$ to be the analogous $k \times n$ matrix for vertices in $[n] \setminus V_1$. Observe that the probability matrix $\mathbf{P}$ can now be represented as $\mathbf{W}_{(1)}^T \mathbf{B} \mathbf{W}_{(2)}$.

The reduction to the Latent Simplex Model can now be stated as follows: given a data matrix $\mathbf{A}$ which is the adjacency matrix of the community graph, and the latent variable matrix $\mathbf{P}$, recover the simplex $\mathbf{M} = \mathbf{W}_{(1)}^T \mathbf{B}$. Further, (Airoldi et al., 2008) assumes that each column of $\mathbf{W}_{(2)}$ is picked from the Dirichlet distribution with parameter $1/k$. Combined with tools from random matrix theory (Vershynin, 2010), (Bhattacharyya & Kannan, 2020) (Lemma 7.2) shows that the Proximate Latent Points and Spectrally Bounded assumptions hold for Stochastic Block Models. As for the Significant Singular Values assumption, it is satisfied when $\sigma$ is a small enough polynomial in $k$.

**Justifying Significant Singular Values.**   We give the following further justification for assumption (4) in the supplementary material: a faster algorithm using the assumptions from (Bhattacharyya & Kannan, 2020) would imply an algorithmic breakthrough for spectral low-rank approximation and partially resolve the first open question of (Woodruff, 2014).

**Theorem 2.2** (Spectral LRA and Learning a Simplex (informal)). *There exists a distribution over instances such that learning a latent simplex in $o(nnz(\mathbf{A}) \cdot k)$ time with good probability implies a constant factor spectral low-rank approximation algorithm in the same running time.*

## 3   ALGORITHM AND ANALYSIS

**Preliminaries.**   We use $n, d,$ and $k$ to denote the number of data points, the number of dimensions of the space and the number of vertices of $\mathcal{K}$ respectively. We use the notation $\mathbf{A}_{*,j}$ to denote the $j$-th column of matrix $\mathbf{A}$. For $\mathbf{A} \in \mathbb{R}^{d \times n}$ with rank $r$, its singular value decomposition, denoted by $\text{SVD}(\mathbf{A}) = \mathbf{U}\boldsymbol{\Sigma}\mathbf{V}^T$, guarantees that $\mathbf{U}$ is a $d \times r$ matrix with orthonormal columns, $\mathbf{V}^T$ is an $r \times n$ matrix with orthonormal rows and $\boldsymbol{\Sigma}$ is an $r \times r$ diagonal matrix. The diagonal entries of $\boldsymbol{\Sigma}$ are the singular values of $\mathbf{A}$, denoted by $\sigma_1 \geq \sigma_2 \geq \ldots \geq \sigma_r$. Given an integer $k \leq r$, we define the truncated singular value decomposition of $\mathbf{A}$ that zeros out all but the top $k$ singular values of $\mathbf{A}$, i.e., $\mathbf{A}_k = \mathbf{U}\boldsymbol{\Sigma}_k\mathbf{V}^T$, where $\boldsymbol{\Sigma}_k$ has only $k$ non-zero entries along the diagonal. It is well-known that the truncated SVD computes the best rank-$k$ approximation to $\mathbf{A}$ under the Frobenius norm, i.e., $\mathbf{A}_k = \min_{\text{rank}(\mathbf{X}) \leq k} \|\mathbf{A} - \mathbf{X}\|_F$. Given an orthonormal basis $\mathbf{U}$ for a subspace, we use $\mathbf{P}_{\mathbf{U}} = \mathbf{U}\mathbf{U}^T$ to denote the projection matrix corresponding to the subspace. We consider the following notion of subspace distance:

**Definition 3.1** ($\sin\Theta$ Distance).   For any two subspaces $\mathbf{R}, \mathbf{S}$ of $\mathbb{R}^d$, the $\sin\Theta$ distance between $\mathbf{R}$ and $\mathbf{S}$ is defined as $\sin\Theta(\mathbf{R}, \mathbf{S}) = \max_{u \in \mathbf{R}} \min_{v \in \mathbf{S}} \sin\theta(u, v) = \max_{u \in \mathbf{R}, |u|=1} \min_{v \in \mathbf{S}} \|u - v\|$.

We use the notion of spectral low-rank approximation to obtain a compact representation of the input and compute matrix-vector products efficiently. We also require the notion of mixed spectral-Frobenius low-rank approximation. This guarantee is weaker than spectral-low rank approximation but admits faster algorithms.

**Definition 3.2** (Spectral Low-rank Approximation, Spectral-Frobenius Low-rank Approximation). Given a matrix $\mathbf{A}$, an integer $k$ and $\epsilon > 0$, a rank-$k$ matrix $\mathbf{B}$ satisfies a *relative-error* spectral low-rank approximation guarantee if $\|\mathbf{A} - \mathbf{B}\|_2^2 \leq (1 + \epsilon)\|\mathbf{A} - \mathbf{A}_k\|_2^2$. $\mathbf{B}$ satisfies a mixed spectral-Frobenius low-rank approximation guarantee if $\|\mathbf{A} - \mathbf{B}\|_2^2 \leq (1 + \epsilon)\|\mathbf{A} - \mathbf{A}_k\|_2^2 + \frac{\epsilon}{k}\|\mathbf{A} - \mathbf{A}_k\|_F^2$.

### 3.1   OVERVIEW

In this section, we provide an overview of our algorithmic techniques and discuss the main challenges we overcome to obtain an input-sparsity time algorithm.

**Our Techniques.**   The starting point in Bhattacharyya & Kannan (2020) is that the smoothened polytope, obtained by averaging points in the data matrix $\mathbf{A}$ is itself close to the latent points in the convex hull of $\mathcal{K}$ in operator norm. This fact is captured by the following lemma:

**Lemma 3.3** (Subset Smoothing). *For any $\mathcal{S} \subset [n]$, let $\mathbf{A}_\mathcal{S}$ be a vector obtained by averaging the columns of $\mathbf{A}$ indexed by $\mathcal{S}$ and define $\mathbf{P}_\mathcal{S}$ similarly. Then for $\|\mathbf{A} - \mathbf{P}\|_2 \leq \sigma\sqrt{n}$, we have $\|\mathbf{A}_\mathcal{S} - \mathbf{P}_\mathcal{S}\|_2 \leq \sigma\sqrt{n/|\mathcal{S}|}$.*

Our main insight is that we can approximately optimize a linear function on the smoothed polytope by working with a rank-$k$ spectral approximation to $\mathbf{A}$ instead. Geometrically, this implies that while the smoothed polytope is perhaps $d$-dimensional, projecting it onto the $k$-dimensional space spanned by the top-$k$ singular values of the data matrix $\mathbf{A}$ suffices to recover the latent $k$-simplex, $\mathcal{K}$. This is surprising since the data matrix can contain points significantly far from the latent polytope. Further, this approach presents several challenges: we do not have access to the left singular space of $\mathbf{A}$ and even if we are provided this subspace exactly, it is unclear why it spans a set of points that approximate vertices of $\mathcal{K}$. Finally, the points obtained by smoothing the projected polytope have no immediate relation to points in the smoothed high-dimensional polytope considered by (Bhattacharyya & Kannan, 2020).

We would like to begin by computing a spectral low-rank approximation (Definition 3.2) for $\mathbf{A}$. Since a low-rank approximation to $\mathbf{A}$ can be represented in factored form $\mathbf{YZ}^T$, where $\mathbf{Y}$ is $d \times k$ and $\mathbf{Z}^T$ is $k \times n$, any matrix-vector product of the form $\mathbf{YZ}^T \cdot x$ only requires $(n + d)k$ time. Thus optimizing a linear function $k$ times over a smoothed low-rank polytope requires only $(n + d)k^2$ time, circumventing the previous bound of $k \cdot \mathsf{nnz}(\mathbf{A})$. However, the best known algorithm for spectral low-rank approximation (Theorem 1 in (Musco & Musco, 2015)) requires $\widetilde{O}(\mathsf{nnz}(\mathbf{A}) \cdot k/\sqrt{\epsilon})$ time and thus provides no improvement. A natural direction to pursue is then to compute a Frobenius low-rank approximation (which requires $\mathsf{nnz}(\mathbf{A})$ time) for $\mathbf{A}$ and use this as our proxy. However, a Frobenius low-rank approximation is too coarse to obtain a subspace that is close to the top-$k$ singular vectors of $\mathbf{A}$.

---

**Algorithm 1 : Learning a Latent $k$-Simplex in Input Sparsity Time**

**Input**: A matrix $\mathbf{A} \in \mathbb{R}^{d \times n}$, integer $k$, and $\epsilon > 0$.

1. Using the algorithm from Lemma 3.4, compute rank-$k$ matrices $\mathbf{Y}, \mathbf{Z}$ such that $\mathbf{YZ}^T$ is a spectral low-rank approximation to $\mathbf{A}$, i.e., $\|\mathbf{A} - \mathbf{YZ}^T\|_2^2 \leq (1 + \epsilon)\|\mathbf{A} - \mathbf{A}_k\|_2^2$.

2. Let $\mathcal{S} = \{\emptyset\}$. For each $t \in [k]$,

   (a) Let $\mathbf{U}_t$ be an orthonormal basis for the vectors in $\mathcal{S}$.

   (b) Compute the projection matrix $\mathbf{P}_t = \mathbf{U}_t\mathbf{U}_t^T$ that projects onto the row span of $\mathcal{S}$.

   (c) Let $g \sim \mathcal{N}(0, \mathbf{I}_k)$ and let $\mathbf{u}_t = g\mathbf{Y}^T(\mathbf{I}_d - \mathbf{P}_t)\mathbf{YZ}^T$ be a random vector in $\mathbb{R}^n$. Compute $\mathcal{R}_t \subset [n]$, a subset of $\delta n$ indices corresponding to the largest coordinates of $\mathbf{u}_t$ in absolute value.

   (d) Let $\mathbf{A}_{\mathcal{R}_t}$ be the average of the columns of $\mathbf{A}$ indexed by $\mathcal{R}_t$. Update $\mathcal{S} = \mathcal{S} \cup \mathbf{A}_{R_t}$.

**Output:** The set of vectors $\mathbf{A}_{\mathcal{R}_1}, \mathbf{A}_{\mathcal{R}_2}, \ldots, \mathbf{A}_{\mathcal{R}_k}$ as our approximation to the vertices of the latent $k$-simplex $\mathcal{K}$.

---

Instead we compute a mixed spectral-Frobenius low-rank approximation (see Definition 3.2) that runs in $O(\mathsf{nnz}(A) + dk^2)$ time, but the resulting error guarantee is weaker. In particular, it incurs an additive $\epsilon\|\mathbf{A} - \mathbf{A}_k\|_F^2/k$ term. Here, we use the assumption we introduced (the Significant Singular Value assumption) to show that the low-rank matrix obtained from this algorithm also satisfies a *relative-error* spectral low-rank approximation guarantee. The next challenge is that the aforementioned guarantee only bounds the spectral norm of $\mathbf{A} - \mathbf{YZ}^T$ in terms of the $(k + 1)$-st singular value of $\mathbf{A}$. This guarantee does not relate how close the subspaces spanned by the columns and rows of the low-rank approximation are to the top-$k$ singular space of $\mathbf{A}$.

A key technical contribution of our work is thus to prove that the subspaces obtained via spectral low-rank approximation are close to the true left and right top-$k$ singular space in angular $(\sin \Theta)$ distance. We note that such a guarantee is crucial to approximately optimize a linear function over $\mathbf{A}$. Further, this result provides an intriguing connection between spectral low-rank approximation and power iteration. It is well known that power iteration suffices to obtain a subspace that is close to the top-$k$ subspace of a matrix in $\sin \Theta$ distance, which at first glance appears much stronger than spectral low-rank approximation. However, our work implies that it suffices to compute a spectral low-rank approximation, which provides a succinct representation of the data matrix and can be computed faster than power iteration in several natural settings.

In the context of learning the latent simplex, given a spectral low-rank approximation, $\mathbf{YZ}^T$, we first restrict to the column span of $\mathbf{Y}$, which w.l.o.g. has orthonormal columns, and iteratively generate $k$ vectors in this subspace. In the first iteration, we generate a random vector $g\mathbf{Y}^T$ and compute $g\mathbf{Y}^T\mathbf{YZ}^T$. We then consider the largest $\delta n$ indices of $g\mathbf{Y}^T\mathbf{YZ}^T$. While the resulting vector does not have strong provable guarantees, we show that averaging the columns of $\mathbf{A}$ corresponding to these indices results in a vector, $\mathbf{A}_{\mathcal{R}_1}$, which intuitively corresponds to efficiently optimizing a linear function over a low-rank approximation to the smoothened polytope, where the smoothened polytope is obtained by averaging over all subsets of $\delta n$ data points. Our next contribution is to show that $\mathbf{A}_{\mathcal{R}_1}$ obtained by the aforementioned algorithmic process is indeed close to a vertex of $\mathcal{K}$.

To obtain an approximation to the remaining vertices of $\mathcal{K}$, we consider the following iterative process: in the $t$-th iteration, consider the subspace $\mathbf{Y}^T(\mathbf{I} - \mathbf{P}_t)$, where $(\mathbf{I} - \mathbf{P}_t)$ is the projection onto the orthogonal complement of the span of $\mathbf{A}_{\mathcal{R}_1}, \mathbf{A}_{\mathcal{R}_2} \ldots \mathbf{A}_{\mathcal{R}_{t-1}}$. Then generate a random vector $g\mathbf{Y}^T(\mathbf{I} - \mathbf{P}_t)$, and compute the largest $\delta n$ coordinates of $g\mathbf{Y}^T(\mathbf{I} - \mathbf{P}_t)\mathbf{Y}\mathbf{Z}^T$. Average the corresponding columns of $\mathbf{A}$ to obtain $\mathbf{A}_{\mathcal{R}_t}$ and output this vector. We prove that after iterating $k$ times, the vectors $\mathbf{A}_{\mathcal{R}_1}, \mathbf{A}_{\mathcal{R}_1}, \ldots \mathbf{A}_{\mathcal{R}_k}$ approximate all the vertices of the latent simplex $\mathcal{K}$ within the desired accuracy and running time.

In contrast, prior work of (Bhattacharyya & Kannan, 2020) uses power iteration to approximate the left top-$k$ singular space $\mathbf{U}_k$ of $\mathbf{A}$ using a subspace $\widehat{\mathbf{V}}$ that is $\text{poly}(\alpha/k)$ close in $\sin\Theta$ distance. Each step of the power iteration uses $O(\text{nnz}(\mathbf{A}) + dk^2)$ time and is repeated $\log(d)$ times. Next, they pick a random vector $u_1$ in the subspace spanned $\widehat{\mathbf{V}}$ and compute $\mathbf{A}_{\mathcal{R}_1} = \text{argmax}_{\mathcal{S}:|\mathcal{S}|=\delta n} |u_1 \cdot \mathbf{A}_{\mathcal{S}}|$, using the resulting vector as an approximation to some vertex $\mathbf{M}_{*,1}$.

They then repeat the above algorithm $k$ times and in the $i$-th iteration, they pick $u_i$ to be a uniformly random direction in the $k-i$ dimensional subspace constructed as follows: let $\widetilde{\mathbf{V}}_{i-1}$ be an orthonormal basis for $\mathbf{A}_{\mathcal{R}_1}, \mathbf{A}_{\mathcal{R}_2}, \ldots, \mathbf{A}_{\mathcal{R}_{i-1}}$. Intuitively, this corresponds to sampling a random vector from the subspace orthogonal to the set of vertex approximations picked thus far. The resulting $k$ vectors $\mathbf{A}_{\mathcal{R}_1}, \ldots, \mathbf{A}_{\mathcal{R}_k}$ are the approximation to the vertices of the latent simplex. Since they directly optimize over the smoothened polytope, the correctness analysis is more straightforward.

However, each iteration of the algorithm requires optimizing a linear function over the smoothened polytope and in particular requires computing $u_i \cdot \mathbf{A}$, and thus, the overall running time is dominated by $k \cdot \text{nnz}(\mathbf{A})$. Since the latent simplex satisfies the Well-Separateness condition, the inner product with a random direction is maximized by a unique vertex. Intuitively, it appears necessary to project away from the set of vectors obtained up to the $i$-th iteration in order to learn new vertices of $\mathcal{K}$. The inherently iterative nature of the algorithm combined with matrix-vector product lower bounds indicates that the new algorithmic ideas we introduce are in fact necessary.

## 3.2 Technical Discussion

In this section, we provide an outline of our proof. We defer the full proofs to the supplementary material. We start with a spectral low-rank approximation for $\mathbf{A}$. We then use the right factor as an approximation to $\mathbf{\Sigma}_k \mathbf{V}_k^T$ and the left factor as an approximation to $\mathbf{U}_k$.

**Lemma 3.4.** *(Input-Sparsity Spectral LRA (Cohen et al., 2015; 2017).) Given a matrix $\mathbf{A} \in \mathbb{R}^{d \times n}$, $\epsilon, \delta > 0$ and $k \in \mathbb{N}$, there exists an algorithm that outputs matrices $\mathbf{Y}, \mathbf{Z}$, such that with probability at least $1 - \delta$, $\|\mathbf{A} - \mathbf{Y}\mathbf{Z}^T\|_2^2 \leq (1 + \epsilon)\|\mathbf{A} - \mathbf{A}_k\|_2^2 + \frac{\epsilon}{k}\|\mathbf{A} - \mathbf{A}_k\|_F^2$, in time $\widetilde{O}\left(\text{nnz}(\mathbf{A}) + (n + d)\text{poly}(k/\epsilon\delta)\right)$.*

Under the Significant Singular Values condition (4), setting $\epsilon = \phi$ in Lemma 3.4 implies with probability $99/100$, $\frac{1}{\text{poly}(k)} \sum_{i=k+1}^n \sigma_i^2 = \frac{1}{\text{poly}(k)}\|\mathbf{A} - \mathbf{A}_k\|_F^2 \leq \sigma_{k+1}^2 = \|\mathbf{A} - \mathbf{A}_k\|_2^2$ and thus $\|\mathbf{A} - \mathbf{Y}\mathbf{Z}^T\|_2^2 \leq 2\|\mathbf{A} - \mathbf{A}_k\|_2^2$. Further, such a matrix $\mathbf{Y}\mathbf{Z}^T$ can be computed in $\widetilde{O}\left(\text{nnz}(\mathbf{A}) + (n + d)\text{poly}(k/\phi)\right)$ time. Next, we show that if $\mathbf{Y}\mathbf{Z}^T$ is a good rank $k$ spectral approximation to $\mathbf{A}$, then the subspace spanned by the columns of $\mathbf{Y}$ must be close to the column span of $\mathbf{U}_k$, the top-$k$ left singular vectors of $\mathbf{A}$. We begin by recalling Wedin's $\sin\Theta$ theorem that relates norms of projectors to angular distance:

**Theorem 3.5** ($\sin\Theta$ theorem (Wedin, 1972)). *Let $\mathbf{R}, \mathbf{S} \in \mathbb{R}^{d \times n}$ and $0 < m \leq \ell$ be integers. Let $\mathbf{R}_m$ and $\mathcal{S}\sigma_\ell$ denote the subspaces spanned by the top $m$ singular vectors of $\mathbf{R}$ and top $\ell$ singular vectors of $\mathbf{S}$, respectively. Suppose $\gamma = \sigma_m(\mathbf{R}) - \sigma_{\ell+1}(\mathbf{S})$. Then, $\sin\Theta(\mathbf{R}_m, \mathcal{S}\sigma_\ell) \leq \frac{\|\mathbf{R}-\mathbf{S}\|_2}{\gamma}$.*

We use the above theorem to bound the spectral norm of the projectors onto the relevant subspaces.

**Lemma 3.6** (Proximity of Subspace Projections). *Let $\mathbf{Y}$ be defined as in Algorithm 1 and let $\mathbf{U}_k$ be the subspace spanned by the top $k$ left singular vectors of $\mathbf{A}$. Let $\mathbf{P_Y}$ and $\mathbf{P_{U_k}}$ be the $d \times d$ projection matrices onto the row span of $\mathbf{Y}$ and $\mathbf{U}_k$. Then $\|\mathbf{P_Y} - \mathbf{P_{U_k}}\|_2 \leq \frac{1}{1000k^{10}}$.*

*Proof Sketch.* Suppose by way of contradiction that $\|\mathbf{P_Y} - \mathbf{P_{U_k}}\|_2 \geq \frac{1}{1000k^{10}}$. This implies $\|\mathbf{U}_k\mathbf{U}_k^T - \mathbf{Y}\mathbf{Y}^T\|_F^2$ is large and thus $\|\mathbf{U}_k\mathbf{Y}^T\|_F^2$ is bounded by $k - \frac{1}{(1000k^{10})^2}$. Intuitively, we

show that if the inner product term is small, we can obtain a lower bound on $\|\mathbf{A} - \mathbf{P_Y}\mathbf{A}\|_2$, as follows: an upper bound on $\|\mathbf{U}_k\mathbf{Y}^T\|_F^2$ suffices to obtain an upper bound on the the $k$-th singular value of $\mathbf{U}_k\mathbf{Y}\mathbf{Y}^T\mathbf{U}^T$ via averaging. This, in turn lower bounds $\|\mathbf{A} - \mathbf{P_Y}\mathbf{A}\|_2 = \|\mathbf{U}^T\Sigma - \mathbf{Y}\mathbf{Y}^T\mathbf{U}^T\Sigma\|_2$ by $\frac{1}{(1000k^{10})^2}\sigma_k(\mathbf{A})$, contradicting the Significant Singular Value assumption. $\square$

Our analysis proceeds via induction on the number of iterations performed by the algorithm. Suppose our algorithm has selected $t$ points from our approximation of the top $k$ subspace and these points are reasonably close to $i$ points of the $k$-simplex. In the $(t + 1)$-st iteration, we again bound the $\sin\Theta$ distance between $\mathbf{Y}^T(\mathbf{I} - \mathbf{P}_t)$, which corresponds to our approximation of the top $k$ subspace projected away from the selected vectors, and the actual $k$-simplex projected away from the corresponding points closest to our selected vectors. This argues that we can continue selecting random vectors in the subspace spanned by $\mathbf{Y}^T(\mathbf{I} - \mathbf{P}_t)$ as a close approximation to random vectors in $\mathbf{M}(\mathbf{I} - \mathbf{P}_t)$. We first bound the $k$-th singular values of the simplex vertices ($\mathbf{M}$) and latent variables ($\mathbf{P}$):

**Lemma 3.7.** *(Bhattacharyya & Kannan, 2020) If the underlying points $\mathbf{M}$ follow the Well-Separateness and Spectrally Bounded Perturbations assumptions, then $\sigma_k(\mathbf{M}) \geq \frac{1000k^{8.5}}{\alpha^2}\frac{\sigma}{\sqrt{\delta}}$, and $\sigma_k(\mathbf{P}) \geq \frac{995k^{8.5}\sqrt{n}}{\alpha^2}\sigma$.*

Next, we prove our lemma relating angular distance of the subspace obtained in the $i$-th iteration of the algorithm ($\mathbf{Y}(\mathbf{I} - \mathbf{P}_i)$) to the optimal subspace ($\mathbf{M}(\mathbf{I} - \mathbf{P}_i)$).

**Lemma 3.8** (Angular Distance between Subspaces.)**.** *For some $r \in [k]$, let $\widehat{\mathbf{M}} = \mathbf{M}_{*,\ell_1} \circ \ldots \circ \mathbf{M}_{*,\ell_r}$ be the matrix with $r$ columns corresponding to vertices of the latent $k$-simplex $\mathbf{M}$ closest to the first $r$ points selected by Algorithm 1, $\mathbf{A}_{\mathcal{R}_1}, \ldots, \mathbf{A}_{\mathcal{R}_r}$, respectively. Suppose $\|\mathbf{A}_{\mathcal{R}_i} - \mathbf{M}_{*,\ell_i}\|_2 \leq \frac{300k^4}{\alpha}\frac{\sigma}{\sqrt{\delta}}$ for each $i \in [r]$. Let $\mathbf{P}_r$ be the projection matrix orthogonal to $\mathbf{A}_{\mathcal{R}_1}, \ldots, \mathbf{A}_{\mathcal{R}_r}$. Then $\sin\Theta\left(\mathbf{Y}(\mathbf{I}_d - \mathbf{P}_r), \textbf{Span}\,(\mathbf{M}) \cap Null(\widehat{\mathbf{M}})\right) \leq \alpha/100k^4$ and $\sin\Theta\left(\textbf{Span}\,(\mathbf{M}) \cap Null(\widehat{\mathbf{M}}), \mathbf{Y}(\mathbf{I}_d - \mathbf{P}_r)\right) \leq \alpha/100k^4$.*

*Proof Sketch.* Let $\mathbf{y} \in \mathbf{Y}(\mathbf{I}_d - \mathbf{P}_r)$ be a unit vector. It can be shown that using the $\sin\Theta$ theorem and the hypothesis that there exists $\mathbf{x} \in \textbf{Span}\,(\mathbf{M})$ with $\|\mathbf{x} - \mathbf{y}\|_2 \leq \frac{\alpha^2}{500k^{8.5}}$. Let $\mathbf{z} = \mathbf{x} - \widehat{\mathbf{M}}\widehat{\mathbf{M}}^\dagger\mathbf{x}$ be the component of $\mathbf{x}$ in Null($\widehat{\mathbf{M}}$). We can then bound $\|\mathbf{x} - \mathbf{z}\|_2 \leq \|\mathbf{x} - \mathbf{y}\|_2 + \|\widehat{\mathbf{M}}(\widehat{\mathbf{M}}^T\widehat{\mathbf{M}})^{-1}(\widehat{\mathbf{M}}^T - \widehat{\mathbf{A}}^T)\mathbf{y}\|_2$, where $\widehat{\mathbf{A}}$ is the set of vectors selected thus far and $\widehat{\mathbf{A}}^T\mathbf{y} = 0$. Combining the aforementioned observations, we can bound $\|\mathbf{x} - \mathbf{z}\|_2$ by $\alpha^2/(500k^{8.5}) + k^{4.5}\sigma/(\alpha\sqrt{\delta}\sigma_k(\widehat{\mathbf{M}}))$. We then observe that $\mathbf{y} \in \mathbf{Y}(\mathbf{I}_d - \mathbf{P}_r)$ and $\mathbf{z} \in \textbf{Span}\,(\mathbf{M}) \cap \text{Null}(\widehat{\mathbf{M}})$, and appeal to Lemma 10.1 in (Bhattacharyya & Kannan, 2020) to yield $\sin\Theta\left(\mathbf{Y}(\mathbf{I}_d - \mathbf{P}_r), \textbf{Span}\,(\mathbf{M}) \cap \text{Null}(\widehat{\mathbf{M}})\right) \leq \alpha/100k^4$.

To prove the second half of the claim, it suffices to show that the dimension of $\mathbf{Y}(\mathbf{I}_d - \mathbf{P}_r)$ is $k - r$, since $\textbf{Span}\,(\mathbf{M}) \cap \text{Null}(\widehat{\mathbf{M}})$ has dimension $k - r$ and the $\sin\Theta$ distance is symmetric between two subspaces of the same dimension. By construction $\mathbf{Y}$ has dimension $k$ so that $\mathbf{Y}(\mathbf{I}_d - \mathbf{P}_r)$ has dimension at least $k - r$. Therefore, there exist vectors $\mathbf{u}_1, \ldots, \mathbf{u}_{k-r+1} \in \mathbf{Y}(\mathbf{I}_d - \mathbf{P}_r)$ and vectors $\mathbf{v}_1, \ldots, \mathbf{v}_{k-r+1} \in \textbf{Span}\,(\mathbf{M}) \cap \text{Null}(\widehat{\mathbf{M}})$ such that $\|\mathbf{u}_i - \mathbf{v}_j\|_2 < \alpha/100k^4$. We can then upper and lower bound $|\mathbf{v}_a \cdot \mathbf{v}_b|$ for all $a \neq b$ to conclude that $\mathbf{v}_1, \ldots, \mathbf{v}_{k-r+1}$ are orthogonal, contradicting that $\textbf{Span}\,(\mathbf{M}) \cap \text{Null}(\widehat{\mathbf{M}})$ spans a $k - r$ dimensional space and the claim follows. $\square$

Now we need to show that our algorithm is (1) well-defined and (2) preserves the invariant that the $(i + 1)$-st point sampled from $\mathbf{Y}^T(\mathbf{I} - \mathbf{P}_i)$ will also be reasonably close to some different point of the $k$-simplex. We show the selected procedure is well-defined in Lemma 3.9 by arguing that there exists a unique solution to the maximization problem.

**Lemma 3.9** (Optimization is Well-Defined)**.** *Let $\mathbf{u} \in \mathbb{R}^d$ be a random unit vector in the space of $\mathbf{Y}^T(\mathbf{I}_d - \mathbf{P}_r)$, where $\mathbf{P}_r$ is the orthogonal projection to $\mathbf{A}_{\mathcal{R}_1}, \ldots, \mathbf{A}_{\mathcal{R}_r}$. Then there exists a constant $c > 0$ so that with probability at least $1 - c/k^{1.5}$:*

*1. For all distinct $a, b \notin \{\ell_1, \ldots, \ell_r\}$, then $|\mathbf{u} \cdot (\mathbf{M}_{*,a} - \mathbf{M}_{*,b})| \geq \frac{0.097}{k^4}\alpha \max_\ell \|\mathbf{M}_{*,\ell}\|_2$.*

2. *For all $a \notin \{\ell_1, \ldots, \ell_r\}$, then $|\mathbf{u} \cdot \mathbf{M}_{*,a}| \geq \frac{0.0989}{k^4} \alpha \max_\ell \|\mathbf{M}_{*,\ell}\|_2$.*

We then show that the algorithm preserves the aforementioned invariant by showing that the unique solution $\mathbf{A}_{\mathcal{R}_i}$ cannot correspond to one of the vertices of the $k$-simplex that have been found in the first $i$ rounds, thus proving that we find a solution $\mathbf{A}_{\mathcal{R}_i}$ that corresponds to a new vertex of $\mathbf{M}$. We then show $\mathbf{A}_{\mathcal{R}_i}$ is close to the new vertex of $\mathbf{M}$, preserving the inductive hypothesis.

**Lemma 3.10** (Recovery Guarantees). *Let $\ell_{r+1} = \mathrm{argmax}_\ell \mathbf{u} \cdot \mathbf{M}_{*,\ell}$ if $\mathbf{u} \cdot \mathbf{A}_{\mathcal{R}_{r+1}} \geq 0$ and be $\mathrm{argmin}_\ell \mathbf{u} \cdot \mathbf{M}_{*,\ell}$ otherwise. Then, $\|\mathbf{A}_{\mathcal{R}_{r+1}} - \mathbf{M}_{*,\ell_{r+1}}\|_2 \leq 300 k^4 \sigma / \alpha \sqrt{\delta}$.*

*Proof Sketch.* We consider the case $\mathbf{u} \cdot \mathbf{A}_{\mathcal{R}_{r+1}} \geq 0$ as the analysis for the case $\mathbf{u} \cdot \mathbf{A}_{\mathcal{R}_{r+1}} < 0$ is symmetric. Let $\ell_{r+1} = \mathrm{argmax}_\ell \mathbf{u} \cdot \mathbf{M}_{*,\ell}$. It can be shown that $\ell_{r+1} \notin \{\ell_1, \ldots, \ell_r\}$. Thus applying Lemma 3.9, $\mathbf{u} \cdot \mathbf{M}_{*,\ell_{r+1}} \geq 0.0989 \alpha \max_\ell \|\mathbf{M}_{*,\ell}\|/k^4$. By the Proximate Latent Points assumption, there exists a set $\sigma_{\ell_{r+1}}$ of size $\delta n$ so that $\|\mathbf{P}_{*,j} - \mathbf{M}_{*,\ell_{r+1}}\|_2 \leq \frac{4\sigma}{\sqrt{\delta}}$ for all $j \in \sigma_{\ell_{r+1}}$ so that $\|\mathbf{P}_{*,\sigma_{\ell_{r+1}}} - \mathbf{M}_{*,\ell_{r+1}}\|_2 \leq \frac{4\sigma}{\sqrt{\delta}}$. Then by Lemma 3.1 in (Bhattacharyya & Kannan, 2020), $\mathbf{u} \cdot \mathbf{A}_{*,\sigma_{\ell_{r+1}}} \geq \mathbf{u} \cdot \mathbf{M}_{*,\ell_{r+1}} - 5\sigma/\sqrt{\delta}$. Similarly, we can show $\mathbf{u} \cdot \mathbf{A}_{\mathcal{R}_{r+1}} \geq \mathbf{u} \cdot \mathbf{M}_{*,\ell_{r+1}} - 8\sigma/\sqrt{\delta}$. Further, using Lemma 3.9 and the Spectrally Bounded Perturbation assumption, we obtain $\mathbf{u} \cdot \mathbf{M}_{*,a} \leq \mathbf{u} \cdot \mathbf{M}_{*,\ell_{r+1}} - 0.097\alpha \max_\ell \|\mathbf{M}_{*,\ell}\|/k^4$. This enables us to upper bound $\mathbf{u} \cdot \mathbf{A}_{\mathcal{R}_{r+1}}$ by $\mathbf{u} \cdot \mathbf{M}_{*,\ell_{r+1}} - 0.097\alpha \max_\ell \|\mathbf{M}_{*,\ell}\|_2 (1 - w_{\ell_{r+1}})/k^4 + 4\sigma/\sqrt{\delta}$. Combining the upper and lower bounds, straightforward computations yield the claim. $\square$

In contrast to (Bhattacharyya & Kannan, 2020), we only need input sparsity time to compute the low-rank approximation to $\mathbf{A}$. The subsequent $k$ iterations of selecting points from $\mathbf{Y}^T(\mathbf{I} - \mathbf{P}_i)$ are computed in the low-dimensional space and use lower-order runtime. Hence, the dominating term in the final runtime is just the input sparsity time used to compute $\mathbf{Y}^T(\mathbf{I} - \mathbf{P}_i)$.

## 4 EMPIRICAL EVALUATION

In this section, we describe a series of experiments that demonstrate the advantage of our algorithm, performed in Python 3.6.9 on an Intel Core i7-8700K 3.70 GHz CPU with 12 cores and 64GB DDR4 memory, using an Nvidia Geforce GTX 1080 Ti 11GB GPU, on both synthetic and real-world data. Whereas previous work requires computing the top $k$ subspace as a pre-processing step, our main improvement is that we only require a crude approximation. Thus we compared the running times for finding the top $k$ subspace as required by (Bhattacharyya & Kannan, 2020) to finding a mixed spectral-Frobenius approximation using an input sparsity algorithm, as required by our algorithm. For the former, we use the `svds` method from the sparse scipy linalg package optimized by LAPACK. For the latter, (Cohen et al., 2015; 2017) show that using a sparse CountSketch matrix (Clarkson & Woodruff, 2013; Meng & Mahoney, 2013; Nelson & Nguyen, 2013), i.e., a matrix with $O(k^2)$ columns and a single nonzero entry in each row that is in a random location and is a random sign, suffices to obtain a mixed spectral-Frobenius guarantee; we evaluate such a matrix with exactly $k^2$ columns. Across all parameters and datasets, the input sparsity procedure used by our algorithm significantly outperforms the optimized power iteration methods required by (Bhattacharyya & Kannan, 2020).

**Synthetic Data.** Since our theoretical results are most interesting when $k \ll d \ll n$, we set $n = 50000$, $d = 1000$, $k \in \{20, 50, 100\}$ and generate a random $d \times n$ matrix $\mathbf{A}$ that consists of independent entries that are each 1 with probability $p \in \left\{\frac{1}{500}, \frac{1}{2000}, \frac{1}{5000}\right\}$ and 0 with probability $1 - p$. In Figure 4.1, we report the average running time of both algorithms, among 5 independent runs for each choice of $p$ and $k$.

**Social Networks.** We also evaluate the algorithms on the `email-Eu-core` network dataset of interactions across email data between individuals from a large European research institution (Yin et al., 2017; Leskovec et al., 2007) and the `com-Youtube` dataset of friendships on the Youtube social network (Yang & Leskovec, 2015), both accessed through the Stanford Network Analysis Project (SNAP). In the former, there are $n = d = 1005$ nodes in the adjacency matrix over 25571 total edges, forming $k = 42$ communities. In the latter, there are 1134890 nodes with 8385 communities, from which we extract a $d \times n$ matrix with $n = 100000$, $d = 1000$ to represent a bipartite graph, as described in both Section 2.2 and (Bhattacharyya & Kannan, 2020). In Figure 4.2, we report the

| Mean Runtime of Algorithms across Parameters | $p = 1/500$ | $p = 1/2000$ | $p = 1/5000$ |
|---|---|---|---|
| Top $k$ Subspace, $k = 20$ | 35.056s | 29.725s | 16.45s |
| Input Sparsity Approximation, $k = 20$ | 0.595s | 0.329s | 0.83s |
| Top $k$ Subspace, $k = 50$ | 56.146s | 54.613s | 53.213s |
| Input Sparsity Approximation, $k = 50$ | 0.658s | 0.657s | 0.434s |
| Top $k$ Subspace, $k = 100$ | 78.420s | 79.410s | 71.424s |
| Input Sparsity Approximation, $k = 100$ | 0.501s | 0.387s | 0.440s |

Figure 4.1: Mean runtime comparison of algorithms across parameters on synthetic data.

running time of both algorithms across each dataset among choices of $k \in \{20, 50, 100\}$. We observe that the resulting matrix has sparsity roughly 1000, which is consistent with $p \approx \frac{1}{n}$ and is much less than the sparsity parameters tested in our synthetic data.

| | email-Eu-core network | com-Youtube |
|---|---|---|
| Top $k$ Subspace, $k = 20$ | 0.387s | 5.713s |
| Input Sparsity Approximation, $k = 20$ | 0.005s | 0.379s |
| Top $k$ Subspace, $k = 50$ | 0.556s | 16.711s |
| Input Sparsity Approximation, $k = 50$ | 0.003s | 0.373s |
| Top $k$ Subspace, $k = 100$ | 1.281s | 41.788s |
| Input Sparsity Approximation, $k = 100$ | 0.003s | 0.366s |

Figure 4.2: Mean runtime comparison of algorithms across parameters on real-world data.

Finally, we consider a full end-to-end implementation comparing the runtime and least squares loss of the top $k$ subspace algorithm and our input sparsity approximation algorithm over various ranges of the parameter $k$ and smoothening parameter $\delta n$ on the com-Youtube dataset, from which we randomly extract an $n \times d$ matrix, with $n = 20000$ and $d = 1000$ to represent a bipartite graph. Our results in Figure 4.3 show that our algorithm not only significantly outperforms the top $k$ subspace algorithm in runtime, but also produces solutions with lower least squared loss.

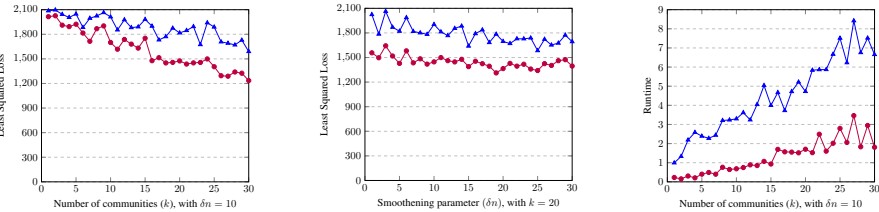

Figure 4.3: Comparison of least squares loss by power iteration algorithm (in blue triangles) and by our algorithm (in red circles), over various ranges of the parameter $k$ with smoothening parameter $\delta n = 10$, and over various ranges of $\delta n$ with $k = 20$, on the com-Youtube dataset. Also runtime comparison over a range of $k$, with $\delta n = 10$.

ACKNOWLEDGMENTS

A.B., and D. W. were supported by the Office of Naval Research (ONR) grant N00014-18-1-2562, and the National Science Foundation (NSF) Grant No. CCF-1815840. D.W and S.Z were supported by National Institute of Health (NIH) grant 5R01 HG 10798-2 and a Simons Investigator Award.

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
