# OpenReview forum: "Learning a Latent Simplex in Input Sparsity Time"
_ICLR.cc/2021/Conference — ICLR 2021 Spotlight_

### Official Review · AnonReviewer3 · 2020-10-29
**Overall, I vote for accepting. The work is tough and technically novel.**

**Rating:** 8
**Confidence:** 3

**Review:**

The paper considers the problem of learning a latent k-vertex simplex K, given a collection A of n points that are obtained by randomly perturbing latent points in the simplex. It improves the prior work [Bhattacharyya and Kannan, SODA 2020] by removing the multiplicative k factor in the running time given an (necessary) assumption about the mass of the top k singular values of A. Empirically, their algorithm is faster than the prior algorithm on both the synthetic and the real-world datasets.

Overall, I vote for accepting. The work is tough and technically novel.

Pros:
    1. The paper provides a detailed discussion on the application of their latent simplex model, which I really appreciate. These discussions make their additional assumption reasonable.
    2. The paper provides a detailed discussion of the technical novelty. Specifically, the idea of obtaining a low-rank approximation to A introduces a small angular distance is interesting and accelerates the running time.

Cons:
    1. The experimental results that your algorithm not only outperforms on the running time but also produces solutions with lower least squared loss are interesting. Could you give some explanation of this phenomenon?

---

> ### Author Response · Authors · 2020-11-25
> **Response to Reviewer 3**
>
> We thank the reviewer for their thoughtful comments on the paper. Although our analysis explains why our algorithm performs better than the previous benchmark in running time, it is true that we do not have any theoretical statements that explain why our algorithm performs better than the previous benchmark in least squares loss. One possible explanation is that a smaller sketch size could perform an implicit form of denoising or regularization for least squares loss; this is an interesting direction to further explore. Another interesting question is which loss functions would be the most appropriate for each application. It could be true that our algorithm does not perform as well as the previous benchmark on specialized loss functions for other datasets.

---

### Official Review · AnonReviewer4 · 2020-10-31
**Very High Quality of Argumentation**

**Rating:** 9
**Confidence:** 4

**Review:**

This is a paper in theoretical computer science which considers what
many would consider a very hard problem -- learning a latent simplex lurking 'underneath' a cloud of data.
It is extremely well organized and tightly put together.

It patiently shows that this problem is lurking behind many different
high-dimensional data analysis challenges, and then carefully lays the groundwork
for discussing a proposed algorithm and  its performance characteristics.

What really surprised me about the paper was that the assumptions
going to make up the 'math result' seemed to be quite stylized to me,
in such a way that many datasets could never really satisfying them
(although perhaps some could). Nevertheless, the authors present results on
real data which seem to suggest that the algorithm really can be used
in empirical representation learning. This really surprises me,
while putting forward positive results based on the authors' own
invented assumptions impresses me not so much.  So it would be interesting
to hear from the authors, for example in a talk or in an eventual journal paper,
to what extent the data actually obey the stated assumptions. If they
don'y obey the stated assumptions of this paper, then some less restrictive
assumptions would be the 'sharp conditions'. In that case, what are the true
conditions needed for the algorithm to work?

Obviously the significance this work will have in the eyes of potential users
depends a lot on the just-discussed  Q and A.

---

> ### Author Response · Authors · 2020-11-25
> **Response to Reviewer 4**
>
> We thank the reviewer for their optimistic assessment of the paper. We agree that understanding which kind of datasets follow our mathematical assumptions would be a valuable contribution for empirical representation learning. It should be noted that our assumptions are data-independent and thus lead to rigourous guarantees whereas in practice, the algorithm can likely succeed with much milder assumptions. It would also be interesting to see whether we could obtain a graceful tradeoff between our assumptions and our theoretical guarantees.
>
> Moreover, it is possible to obtain $\text{nnz}(A) + n\cdot\text{poly}(\text{stable-rank}(A), k, 1/\epsilon)$ time via sketching for spectral low rank approximation. Thus even if our assumption is violated, this sketch gives a tradeoff and may still be better than the previous asymptotically fastest algorithm in the earlier paper if the stable rank is small.

---

### Official Review · AnonReviewer2 · 2020-11-01
**Latent Simplex in Input Sparsity Time**

**Rating:** 7
**Confidence:** 4

**Review:**

The paper considers the problem of learning/finding a simplex with k vertices in R^d from a set of points "close" to the simplex. That is, given a set of points A, and integer k, the goal is to find subsets R_1, ... , R_k of A such that the mean of R_i is close to the ith vertex of the underlying simplex.

This problem is considered by Bhattacharyya and Kannan (SODA 2020) and they gave an O(k nnz(A)) algorithm assuming that the data follows certain assumptions: 1) well-separateness, 2) proximate latent points, and 3) spectrally bounded perturbations.

The current paper adds an additional assumption of significant singular values and presents an O(nnz(A)) algorithm for this case. The paper argues that in some important machine learning applications (e.g., LDA), the newly introduced assumption holds. Moreover, they show that only considering the three assumptions of Bhattacharyya and Kannan, there does not exist an asymptotically faster algorithm than of Bhattacharyya and Kannan (unless there exists a constant factor spectral low-rank approximation algorithm in the same running time). Finally, the paper presents empirical evaluations on synthetic and real-world datasets and shows that the proposed method achieves a better running time and least square error compared to the prior method.

The paper is very well-written. It is very interesting that LDA and stochastic block models satisfy this assumption. Moreover, the analysis of the algorithm is novel and interesting. I am wondering if this method can be used for learning a general V-polytope. How could we change the assumptions so the algorithm would work for a V-polytope? and what would be the running time?

---

> ### Author Response · Authors · 2020-11-25
> **Response to Reviewer 2**
>
> We thank the reviewer for their positive feedback. We agree that the question of learning a general V-polytope is an interesting potential application. In the most general setting, without any assumption on the input distribution, the convex hull can be formed from $n$ points in general position. Accordingly, the parameter $k$ in our setting becomes roughly $O(n)$ so that the previously lower order $\text{poly}(k)$ terms are no longer negligible.
>
> However, if the convex hull can be approximated by a small number of points, then the problem of learning the V-polytope reduces exactly to our setting, even if some points of the V-polytope are adversarially perturbed. Thus our algorithm can indeed learn a general V-polytope under this input distribution assumption, even in the presence of noise.

---

### Decision · Program_Chairs · 2021-01-07
**Final Decision**

**Decision:**

Accept (Spotlight)

**Comment:**


This paper considers the problem of learning a k-dimensional latent simplices given perturbations of data points in the simplex: this problem is of wide relevance in machine learning as it encompasses many latent variable models including Latent Dirichlet Allocation and Stochastic Block Models. It presents a modification of the recent algorithm of Bhattacharya & Kannan (SODA, 2020) which takes time O(k * nnz(A)), where A is the matrix of perturbed data points. The modified algorithm works with a low-dimensional sketch of the matrix A instead of A, and thereby avoids the dependence on k in the running time of the original algorithm, which used k passes over the data set. The main result of the paper is thus that the latent simplex problem can be solved in O(nnz(A)) time for instances of the problem that satisfy the "Spectrally Bounded Perturbation" property introduced by the authors.

The main questions of the reviewers concerned the question of whether the assumptions needed for the analysis of the novel algorithm to apply hold on real data sets. The authors point out that the assumptions may be stronger than are needed in practice, and suggest that the assumptions could be weakened to assuming that a spectrally-accurate sketch could be used--- this would increase the run-time dependence from just nnz(A), but weakens the assumptions needed. It was also observed that, in addition to a faster runtime, the method outperforms the benchmark method.

This paper should be accepted, due to its theoretical and practical contributions to the problem of latent simplex recovery: it presents an algorithm that provably runs in true input sparsity time given an amenable instance, and practically this algorithms performs well relative to the baseline, verifying the theoretical claims.